# Analysis on Management Policies on Water Quantity Conflict in Transboundary Rivers Embedded with Virtual Water—Using Ili River as the Case

Xia Xu [1], Fengping Wu [2,*], Qianwen Yu [3,*], Xiangnan Chen [2] and Yue Zhao [2]

[1] Architecture and Enginee School, Tongling University, Tongling 244000, China; 027627@tlu.edu.cn
[2] Business School, Hohai University, Nanjing 211000, China; 180213120002@hhu.edu.cn (X.C.); zh_yyyyy@hhu.edu.cn (Y.Z.)
[3] Business School, Suzhou University of Science and Technology, Suzhou 215009, China
* Correspondence: wfp@hhu.edu.cn (F.W.); yuqianwen1992@163.com (Q.Y.)

**Abstract:** Current studies neglect how virtual water transfer (VWT) between countries within a drainage basin affects water stress and then yields an invisible effect on the water quantity conflict in transboundary rivers, which would further make management policies on water quantity conflict less fair and reasonable. Therefore, this study first constructs the Inequality Index of VWT and water stress index (WSI) to assess water stress. Next, different types are set according to the Inequality Index and WSI to analyze management policies, with Ili River as the case. Results show: (1) Within the study period, from 1996, the Inequality Index of VWT between China and Kazakhstan stayed at 0.368 (0.368 < 0.5), indicating a relatively high inequality of VWT between the two countries—China at a disadvantage, while Kazakhstan having the upper hand. (2) According to the remotely sensed data, WSI in the riparian zones of the Ili River rose from 0.288 to 0421 in China, and 0214 to 0.402 in Kazakhstan, showing intermediate scarcity. (3) China and Kazakhstan both fall into Type 2, and Kazakhstan has the advantageous position. Therefore, while allocating the water resources of the Ili River, Kazakhstan should lower its expectation and proactively ask to exchange benefits in other aspects to reverse the outward transfer of its physical water. In addition, the two countries should find suitable ways to go about water rights trading to reduce the possibility of potential water quantity conflict.

**Keywords:** transboundary river; Inequality Index of VWT; WSI; management policies on water quantity conflict

## 1. Introduction

### 1.1. Literature Review

The intensifying global warming and human activity cause the serious scarcity of physical fresh water worldwide. Water stress, in turn, becomes the key trigger of water quantity conflict among countries within the same drainage basin [1–8], and such conflict is the main contributor to water-related conflict in transboundary rivers [1–4]. So, how to alleviate or avoid such conflict is a research topic that needs urgent exploring.

As water quantity conflict is one of the most important tasks for national governments, various attempts have been taken to deal with it. Some scholars proposed to solve the water conflict via international laws and regulations [9–13]. More scholars thought the traditional methods, including economic theories and policy making, could not satisfy the diversified needs of stakeholders and would even intensify the conflict in water resource allocation. Hence, making fair and reasonable allocation rules is key to avoiding such conflict [14–23]. In addition, a few scholars proposed to construct an international water trading market to solve water conflict in transboundary rivers [24–29]. However, current studies mostly focus on the physical water to find solutions to water quantity conflict yet

ignore the virtual water transfer (VWT) triggered by commodity trade among countries within the same drainage basin.

In fact, VWT embedded in trading among countries within a drainage basin affects their water stress—water stress in net importers of virtual water would ease, but that in net exporters would worsen [30–38]. If such virtual water is converted and embedded into the available water in the riparian zones of these countries, it would affect the actual water stress, and yield an invisible effect on the water quantity conflict. Angelis et al. [39] found a correlation between VWT and water conflict among countries within a drainage basin, and water makes the same contributions as petroleum and natural gas do to inter-state conflict. Merz L et al. [40] said that as distant places around the world become increasingly connected through trade, water faces higher stress, and transboundary watershed conflicts have already occurred in some of these basins. Tian and Wang [41] proved that trade of commodities in countries along transboundary rivers would cause more water transfer, affecting the actual amount of water of these countries, leading to inevitable water conflict. All these studies help make clear of the correlation between water quantity conflict and VWT yet contributing little to the relevant management policies. Therefore, we analyze how VWT embedded into physical water rights allocation and proposed management policies, then making the management policies fairer and more reasonable.

Meanwhile, virtual water embedded in trade causes discrepancy between economic benefit and the cost of water resources in countries within the same drainage basin [42,43]. It is, in a word, inequality of VWT, or trade in value added (TiVA). In other words, if to include the virtual water embedded in trade into water quantity issues of transboundary rivers, to look into how VWT affects the water quantity conflict, then the economic benefit from trade (TiVA) among countries within a drainage basin must be considered. On such basis, this study analyzes the Inequality Index of VWT between China and Kazakhstan. Now, several models of inequality are commonly used. Hei [44] first used Gini coefficient to probe into the inequality in the carbon emissions of international trade. After that, scholars applied Gini coefficient, Kakwani measure, Theil index, Lorenz curve, and the coefficient of variation (CV) to study inequality [45–50]. In recent years, scholars have turned to the environmentally extended multiregional input–output (EE-MRIO) model, as it is able to fully demonstrate the complex input–output relationship among different departments or regions and able to track the environmental effect behind commodity trade. To be specific, scholars combined EE-MIRO with the above coefficients to figure out the inequality between TiVA and environmental shifts [51,52]. Most studies using Gini coefficient and Kakwani measure focus on the holistic perspective of trade–environment inequality. However, there are not enough studies using the EE-MRIO model to look into the economy–environment inequality in trade among countries within a drainage basin from a holistic perspective, let alone the dynamic evaluation. Therefore, this study applies the EE-MRIO model and calculates the VWT and transferred TiVA in countries within the same drainage basin to construct the Inequality Index of VWT to analyze the inequality between VWT and TiVA among these countries and propose relevant policies on managing water quantity conflict in transboundary rivers accordingly.

In addition, based on the first paragraph, we can know that basin physical water stress is the key trigger of water quantity conflict among countries within the same drainage basin [1–7]. So, aside from embedding virtual water in the water rights allocation, we also need to combine other strategies based on the water stress types to minimize the physical water pressure of riparian countries in the basin. In this article, virtual water is embedded and combined with physical water stress types into different management policies to reduce water conflict. The current models of assessing water stress mainly include water stress index (the ratio of annual human water use to annual runoff, WSI), criticality ratio (the proportion of water use to total renewable water, CR), and Falkenmark water stress indicator (renewable water appropriated per person per year, FWSI), which are centered around water consumption and water availability [53–62]. Therefore, we use the ratio of water consumption to water availability (i.e., WSI) to measure water stress.

In general, most current policies on managing water quantity conflict in transboundary rivers are only from physical water resources perspective and ignore the invisible effect of VWT. Meanwhile, most studies do not combine VWT with physical water stress types to propose portfolio management policy. So, to fill this gap, this study first uses the EE-MIRO model to calculate the VWT and transferred TiVA, to construct the Inequality Index of VWT. Next, this study establishes WSI to measure water stress in the riparian zones. After that, different types are set according to the Inequality Index and WSI, to probe into the management policies on water quantity conflict in transboundary rivers in different types. Last, with the Ili River as a case, this study probes into how to make fairer and more reasonable policies.

This study makes the following contributions: (1) embedding virtual water transfer into physical water rights allocation and proposing fairer and more reasonable water rights allocation to address transboundary river water quantity conflict issue; (2) combining the inequality of VWT and physical water resources stress to propose more effective policies.

The rest of this paper is structured as follows: Section 2 describes the research framework and study area; Section 3 introduces the models; Section 4 is the major results; Section 5 is the discussion; and Section 6 presents the conclusion.

### 1.2. Research Framework

First of all, based on the MIRO model, this study measures the VWT and transferred TiVA of China and Kazakhstan, constructs a model to calculate the Inequality Index of VWT, assesses the water stress in the riparian zones of both countries based on WSI, then combines Inequality Index of VWT and WSI to find different classifications to optimize management policies. Lastly, this study carries out a case study on the Ili River (Figure 1).

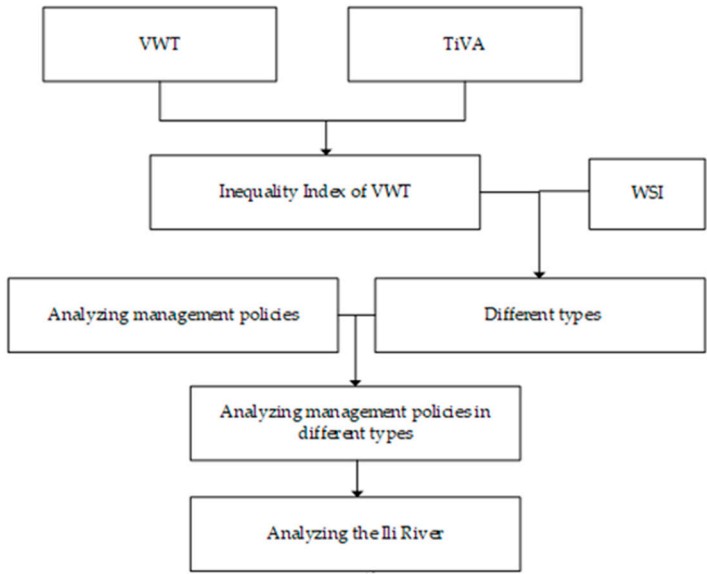

**Figure 1.** Research framework.

Besides, virtual water transfers are at the national level, while after basin countries trade generated virtual water transfer, it is redistributed within the country. It triggers changes in the available water volume of each coastal country in the Ili River Basin. This, in turn, has an intangible impact on water stress in basin coastal countries. Therefore, in this paper, firstly, based on inequity index, we consider embedded VWT into water rights distribution. Then, combining the inequality of VWT and physical water resources stress to propose fairer and more reasonable policies.

## 2. Study Area and Data Sources

*2.1. Overview of Study Area*

2.1.1. Brief Introduction of Ili River

There are a total of 24 transboundary rivers between China and Kazakhstan, of which the Ili River is the most important one and the one with the most prominent water quantity conflict [63]. China is at the upper stream, while Kazakhstan is at lower stream. The riparian zones in both countries are all relatively dry and mainly used for agriculture. In addition, the Ili River covers an area of $15.12 \times 10^4$ km$^2$ with $9.45 \times 10^4$ km$^2$ in Kazakhstan and $5.67 \times 10^4$ km$^2$ in China. In addition, the Ili River contains $228.7 \times 10^8$ m$^3$ of surface water resource, with $58.3 \times 10^8$ m$^3$ in Kazakhstan accounting for 25.5%, and $170.4 \times 10^8$ m$^3$ in China accounting for 74.5% [63]. In addition, firstly, Kazakhstan is the largest landlocked country in the world, and its fresh water resources mainly come from transboundary rivers. In addition, Kazakhstan is generally a relatively arid country, with less water resources than European countries and the Siberian region, thus sensitive to water resources. Secondly, according to our research results, Kazakhstan is a net virtual water exporter in virtual water trade between China and Kazakhstan, and with Kazakhstan as a net virtual water exporter, it lacks water resources, thus forcing it to pay more attention to the water quantity of the Ili River.

2.1.2. Overview of Water Quantity Conflict

The main trigger for the conflict between China and Kazakhstan over the Ili River is lack of water (low water availability and high water consumption). During 1989–2010, the average irrigation requirement in Ili Kazakh Autonomous Prefecture of China rose from 175 mm to 500 mm. In addition, Kazakhstan also needs a huge amount of water for irrigation with an irrigation area of 447,500 hectares from Ili River [64]. Therefore, considering the huge demand for water, combined with the worsening natural environment and rapid economic development, the riparian zones of Ili River in both countries are sensitive to water quantity. For a long time, Xinjiang's increase in water use has raised Kazakhstan's concerns about its water use security. In addition, any fluctuation in the water quantity out of Xinjiang would cause dissatisfaction from Kazakhstan. China asserts that Xinjiang uses much less water than Kazakhstan yet makes greater contributions to the net runoff of the Ili River. Therefore, water quantity conflict between the two countries is centered around who overuses more water of the Ili River [63]. Since the 1990s, China and Kazakhstan have had friendly consultations and negotiations about the water quantity of transboundary rivers, but they have not reached a consensus due to the complicated water allocation and conflicting water needs.

*2.2. Data Source*

(1) Data related to water stress, including annual rainfall, annual evaporation, and water consumption in agriculture, industry, and daily life about the Ili River is collected from relevant literature and databases of NASA and ESA [65–68] for further conversion and calculation. (2) Data on VWT between China and Kazakhstan is collected from EORA website [69]. In addition, global input–output tables are from five sources—EORA, EXIOBASE, Global Trade Analysis Project (GATP), world input–output database (WIOD), and inter-country input–output (ICIO) tables. Given public data about VWT in Kazakhstan in EORA is only available from 1990 to 2015, this study selects 1990–2015 as the study period accordingly.

## 3. Model

The model in this paper consists of two aspects in order in order to obtain a fairer and more reasonable management policy, the model in this paper consists of two aspects. First, determined inequity index, constructed by VWT and TiVA. In addition, in order to achieve more effective management policies for reduced water quantity conflict, we determined the basin riparian zone's physical water stress. Then, based on water stress

index, we combined different strategies to alleviate water stress and reduce water quantity conflict. Finally, combining physical water resources stress with the virtual water transfers, we proposed more equitable, rational, and effective management policies.

*3.1. Constructing Inequality Index of VWT*

The inequity index measures the equivalence of virtual water transfers and economic benefits between basin countries. For example, if basin country *r* has a net input virtual water (VW) and net output TiVA (gained economic benefits) to basin country s, then it has an absolute advantage. Therefore, basin country *r* needs to consider compensating basin country *s* in the water rights allocation. Conversely, basin country s needs to consider compensating basin country *r* in water rights allocation. In addition, if the basin country *r* has a net output VW and TiVA or a net input VW and TiVA, then it needs to be compared to the global trade fair line and obtained in a relatively advantaged or disadvantaged position, thus consider how to embed virtual water in the water rights allocation.

3.1.1. Measuring Transferred TiVA and VWT

We assume that a transboundary river crosses ($m - 1$) countries within a drainage basin, and order them as country 1, country 2, . . . and country $m - 1$. Other countries outside the basin are denoted as other country m. Then, based on the EE-MRIO table, a modified EE-MRIO table is constructed (See Table A1 in Appendix A).

(1) Model of transferred TiVA

TiVA measures the value added by each country in the production of goods and services for import and export, and transferred TiVA means how much value added is transferred from the gross trade volume [42,43]. The net transfer of TiVA is the export of value minus import of value—positive results mean the country runs a trade surplus, generally gaining economic benefits, whereas negative result means the country runs a trade deficit, losing economic benefits [42,43].

Based on Table A1, the formula is constructed as follows:

$$V^{rs} = \sum_{p=1}^{m} G^r L^{rp} F^{ps} \tag{1}$$

where *r* and *s* refer to countries within a drainage basin, *p* stands for all the countries trading with country *r* and country *s*, *m* is the number of countries trading with country *r* and country *s*. $V^{rs}$ is the transferred TiVA from country *r* to country *s*. $G^r$ means the TiVA of country *r*. *L* stands for the Leontief inverse matrix, and $L^{rp}$ the submatrix of Leontief inverse matrix from country *r* to country *p*. *F* stands for the final demand vector, and $F^{ps}$ the final demand vector from country p to country *s*. $G^r L^{rp} F^{ps}$ means the TiVA in country *r* is driven by the final demand in country *p* and consumed in country *s*. In other words, the TiVA is first transferred from country *r* to country *p* as intermediate goods, processed in country *p*, exported to country *s*, and, at last, consumed in country *s*. When *p* = *r*, the final demand of country *s* directly drives the TiVA of country *r*; when $p \neq r$, the final demand of country *s* indirectly drives the TiVA of country *r*.

The driving force of the final demand of country *r* on the TiVA of country *s* is:

$$V^{sr} = \sum_{p=1}^{m} G^s L^{sp} F^{pr} \tag{2}$$

In addition, the net transferred TiVA from country *r* to country *s* is:

$$kv^{rs} = \sum_{p=1}^{m} V^r L^{rp} F^{ps} - \sum_{p=1}^{m} V^s L^{sp} F^{pr} \tag{3}$$

The net transferred TiVA from country $s$ to country $r$ is:

$$kv^{sr} = \sum_{p=1}^{m} V^s L^{sp} F^{pr} - \sum_{p=1}^{m} V^r L^{rp} F^{ps} \tag{4}$$

(2) Model of VWT

Similarly, VWT between two countries is obtained as follows:

$$Z^{rs} = \sum_{p=1}^{m} W^r L^{rp} F^{ps} \tag{5}$$

where $Z^{rs}$ is the VWT from country $r$ to country $s$. When $p = r$, $Z^{rs}$ is the VWT direct trade from country $r$ to country $s$. When $p \neq r$, $Z^{rs}$ is the indirect VWT from country $r$ to country $s$. $W$ refers to the direct water coefficient matrix. $W^r$ is the direct water coefficient matrix of country $r$.

In addition, the VWT from country $s$ to country $r$ is as follows:

$$Z^{sr} = \sum_{p=1}^{m} W^s L^{sp} F^{pr} \tag{6}$$

where $W^s$ stands for the direct water coefficient matrix of country $s$, $L^{sp}$ is the submatrix of $L$ from country $s$ to country $p$, $F^{pr}$ is the submatrix of the final demand matrix from country $p$ to country $r$.

Therefore, the net VWT from country $r$ to country $s$ is as follows:

$$kz^{rs} = \sum_{p=1}^{m} W^r L^{rp} F^{ps} - \sum_{p=1}^{m} W^s L^{sp} F^{pr} \tag{7}$$

where $kz^{rs}$ is the net VWT from country $r$ to country $s$.

Ditto, the net VWT from country $s$ to country $r$ is as follows:

$$kz^{sr} = \sum_{p=1}^{m} W^s L^{sp} F^{pr} - \sum_{p=1}^{m} W^r L^{rp} F^{ps} \tag{8}$$

3.1.2. Constructing Inequality Index Model

With country r as an example, this study constructs a Cartesian coordinate system—net VWT as the $x$-axis, and net transfer of TiVA as the $y$-axis, to represent the relationship between VWT and net transfer of TiVA of countries within the same drainage basin (Figure 2).

Figure 2 shows two types—same flow direction and opposite flow direction. Therefore, in this study, when the net VWT of country $r$ ($kz^{rs}$) is set positive, the net transfer of TiVA ($kv^{rs}$) can be positive or negative, to form three relations—AA', BB', and CC' (Figure 3). Additionally, countries within a drainage basin may be under three conditions—absolute advantage, absolute disadvantage, or advantage in a certain aspect. Therefore, this study interprets "absolute advantage" and "absolute disadvantage" as "highly unequal VWT," and measures "advantage in a certain aspect" via the Inequality Index of VWT.

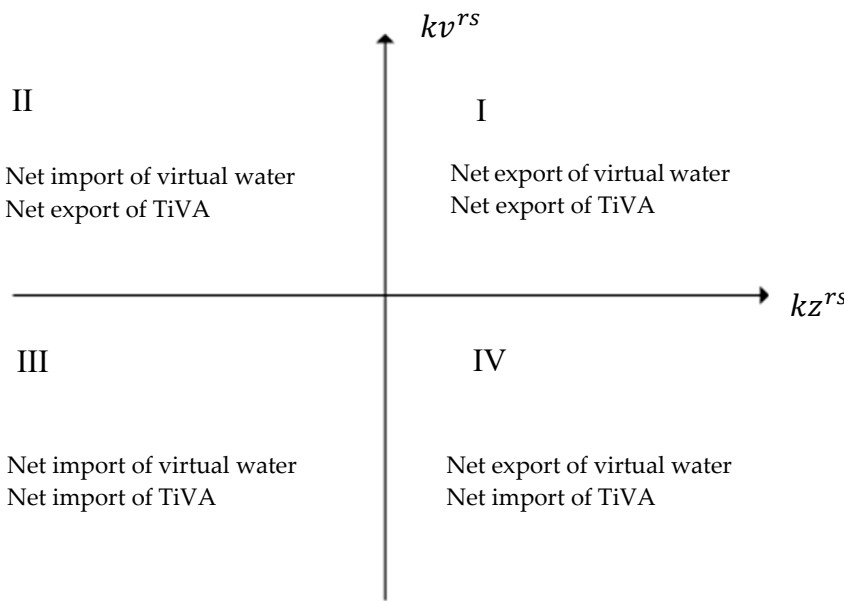

**Figure 2.** Cartesian coordinate system of VWT and net transfer of TiVA.

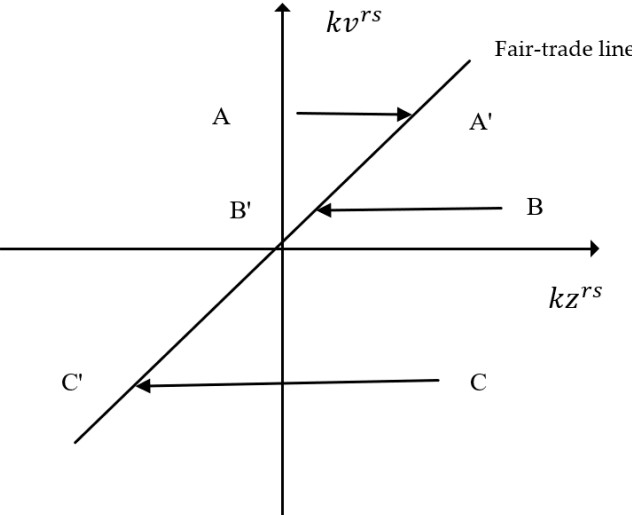

**Figure 3.** Inequality Index of VWT.

Assume the straight line crossing the origin of coordinates in Figure 3 is the fair-trade line, and the dots along the fair-trade line represent the global average level of transferred TiVA–VWT relationship. In other words, the net export (import) of TiVA triggered by net export (import) of VWT is in line with the global average value. From a global perspective, international trade causes the net transfer of virtual water and TiVA, and the transferred TiVA triggered by unit VWT—the slope of the fair-trade line, can be presented as follows:

$$\beta = \sum_{r=1}^{m} \sum_{s=1, r \neq s}^{m} |kv^{rs}| \bigg/ \sum_{r=1}^{m} \sum_{s=1, r \neq s}^{m} |kz^{rs}| \tag{9}$$

where $\beta$ is the slope of the fair-trade line for TiVA–VWT and $\beta \geq 0$; $|kv^{rs}|$ is the absolute value of the transferred TiVA from country $r$ to country $s$; $|kz^{rs}|$ is the absolute value of VWT from country $r$ to country $s$; $\sum_{r=1}^{m} \sum_{s=1, r \neq s}^{m} |kv^{rs}|/2$ is the sum of the net transfer of TiVA of all countries in the world; and $\sum_{r=1}^{m} \sum_{s=1, r \neq s}^{m} |kz^{rs}|/2$ is the sum of the net VWT of all countries in the world.

This study assumes $kz^r > 0$, so the dots can only appear on the right side of the Y-axis in Figure 3—Quadrant I. The dots deviating from the fair-trade line—A, B, and C,

correspond to A′, B′, and C′ on the fair-trade line, and line segments AA′, BB′, and CC′ represent the net VWT that needs to be added/subtracted for the corresponding two dots to reach the fair-trade line. In addition, the deflection distance is defined as DP, to represent the degree of deviation from the fair-trade line:

$$DP = \begin{cases} \frac{\frac{kv^{rs}}{\beta} - kz^{rs}}{\frac{kv^{rs}}{\beta}} = 1 - \frac{\beta kz^{rs}}{kv^{rs}}, \frac{kv^{rs}}{\beta} > kz^{rs} > 0, 0 \le DP < 1 \\ \frac{kz^{rs} - \frac{kv^{rs}}{\beta}}{kz^{rs}} = 1 - \frac{kv^{rs}}{\beta kz^{rs}}, 0 < \frac{kv^{rs}}{\beta} < kz^{rs}, 0 \le DP < 1 \\ \frac{kz^{rs} - \frac{kv^{rs}}{\beta}}{kz^{rs}} = 1 - \frac{kv^{rs}}{\beta kz^{rs}}, \qquad \frac{kv^{rs}}{\beta} < 0, \qquad DP > 1 \end{cases} \qquad (10)$$

where when the transferred TiVA and VWT are located in Quadrant I, *DP* is $[0, 1)$; when they are in Quadrant IV, *DP* is $[1, \infty)$; higher *DP* means higher deviation from the fair-trade line; when *DP* = 0, the dot is on the fair-trade line, representing an equal status.

Therefore, based on *DP*, this study introduces the exponential function $y = e^{-x}$ to construct the Inequality Index of VWT (*Ii*), and unify *Ii* within $(0, 1]$. The details are as follows:

$$Ii = e^{-DP} = \begin{cases} e^{-(1 - \frac{\beta kz^{rs}}{kv^{rs}})}, \frac{kv^{rs}}{\beta} \ge kz^{rs}, \text{Dot above fair} - \text{trade line} \\ e^{-(1 - \frac{kv^{rs}}{\beta kz^{rs}})}, \frac{kv^{rs}}{\beta} \le kz^{rs}, \text{Dot below fair} - \text{trade line} \end{cases} \qquad (11)$$

Wherever the dot is located, *Ii* is always within $(0, 1]$; when *Ii* =1, the dot is on the fair-trade line, indicating an equal status; the closer *Ii* is to 0, the more inequal the VWT is; moreover, the Inequality Index represents the global average level, and is measured in a relatively dynamic manner based on the annual MIRO model.

### 3.2. Model of Water Stress

#### 3.2.1. Indicators to Assess Water Stress

This study uses WSI (the ratio of water consumption to water availability) to measure water stress. According to some scholars. [31,55], annual water availability (annual runoff) equals annual precipitation subtracted by annual evapotranspiration. Annual water consumption includes water for industrial, agricultural, and domestic use. In addition, given that the ecosystem also needs water to provide goods and services for human beings, this study takes ecological water demand into consideration. In addition, desalinated water and imported physical water are also included into the water availability in this study. Indicators to assess water stress without VWT are detailed in Table 1.

**Table 1.** Indicators to assess water stress.

| Research Objective | Indicators | | Variables (m³/year) | Symbol |
|---|---|---|---|---|
| Water stress | Total annual water availability | Annual water availability (annual runoff) | Annual precipitation | $Q_1$ |
| | | | Annual evapotranspiration | $Q_2$ |
| | | Other water availability | Desalinated water | $Q_3$ |
| | | | Imported physical water | $Q_4$ |
| | Total annual water consumption | | Agricultural use | $W_1$ |
| | | | Industrial use | $W_2$ |
| | | | Ecological use | $W_3$ |
| | | | Domestic use | $W_4$ |

#### 3.2.2. Water Stress Assessment Model

(1) Equation to evaluate water stress

Based on Table 1 and the model constructed by Smakhtin et al. [54], we build a model to evaluate water stress. The equation is as follows:

$$X^r = \frac{\sum_{h=1}^{4} W_h^r}{Q_1^r - Q_2^r + Q_3^r + Q_4^r} (r = 1, 2, 3, \cdots, m-1) \tag{12}$$

where $X$ is WSI, $X^r$ is the WSI of riparian areas in country $r$, $W_h^r (h = 1, 2, 3, 4)$ refers to the total annual water use of riparian areas in country $r$, and $Q_1^r - Q_2^r + Q_3^r + Q_4^r$ is the total annual water availability of riparian areas in country $r$.

### 3.2.3. Determining the Indicators of Water Stress

As the annual water availability $(Q_1 - Q_2)$, agricultural $(W_1)$, industrial $(W_2)$, ecological $(W_3)$, and domestic $(W_4)$ water use need to be calculated, this section focuses on determining the estimates of these indicators. Using the country r as an example.

(1) Annual water availability

$$Q_1^r - Q_2^r = (Q_1^{ar} - Q_2^{ar}) \times S^r \times 12 \tag{13}$$

where $Q_1^{ar}$ is average annual precipitation (unit: mm), $Q_2^{ar}$ is average annual evapotranspiration (unit: mm). $Q_1^{ar}$ and $Q_2^{ar}$ are remotely sensed data. They are downloaded through NASA and averaged over administrative areas by Arcgis mask clipping. $S^r$ is the riparian area in country $r$ (unit: km$^2$).

(2) Total annual water consumption

a. Agricultural water use $(W_1^r)$:

$$W_1^r = W_1 \times \frac{S^{br}}{S^b} \tag{14}$$

where $S^{br}$ is the agricultural land area of the riparian areas in country $r$ (unit: km$^2$), and $S^b$ is the total area of agricultural land in country $r$ (unit: km$^2$). $S^{br}$ and $S^b$ are obtained from ESA land use and land cover and then extracted using spatial statistical analysis tools.

b. Industrial and domestic water use $(W_2^r + W_4^r)$. As built-up area includes the area of residential land and industrial land, the amount of domestic and industrial water use in the riparian zone in country r can be measured from built-up area:

$$W_2^r + W_4^r = (W_2 + W_4) \times \frac{S^{cr}}{S^c} \tag{15}$$

where $S^{cr}$ is the built-up area of the riparian areas in country $r$ (unit: km$^2$), and $S^c$ is the total area of built-up area (unit: km$^2$). $S^{cr}$ and $S^c$ are also obtained from ESA land use and land cover and then extracted using spatial statistical analysis tools.

The c represents ecological water use $(W_3^r)$. The data of ecological water use is difficult to obtain. According to Mekonnen and Hoekstra [57], if ecological water use is less than 20% of the available water, it will pose a threat to a river's ecosystem. Therefore, it is assumed that ecological water use in the riparian zone of each country is 20% of the available water.

$$W_3^r = 20\% \times (Q_1^r - Q_2^r) \tag{16}$$

### 3.3. Thresholds of Inequality Index and WSI

Based on Formula (11), the closer $Ii$ gets to 1, the more equal the condition is. Therefore, this study assumes that when $Ii$ is within $(0, 0.5]$, VWT is more inequal; when $Ii$ is within $(0.50, 1.00]$, VWT is less inequal. Next, according to Mekonnen et al. [57], this study classifies the range of $(0, 0.25]$ as low water stress, $(0.25, 0.50]$ medium water stress, and $(0.50, 1.00]$ high water stress. Lastly, six types are found according to $Ii$ and WSI, with country r as the example (Table 2).

**Table 2.** Types based on WSI$^r$ and Inequality Index of VWT (*Ii*).

| Types | *WSI$^r$* and i |
|---|---|
| Type 1 | $0.00 < \text{WSI}^r \le 0.25$, $0.00 < Ii \le 0.05$ |
| Type 2 | $0.25 < \text{WSI}^r \le 0.50$, $0.00 < Ii \le 0.05$ |
| Type 3 | $0.50 < \text{WSI}^r \le 1.00$, $0.00 < Ii \le 0.05$ |
| Type 4 | $0.00 < \text{WSI}^r \le 0.25$, $0.05 < Ii \le 1.00$ |
| Type 5 | $0.25 < \text{WSI}^r \le 0.50$, $0.05 < Ii \le 1.00$ |
| Type 6 | $0.50 < \text{WSI}^r \le 1.00$, $0.05 < Ii \le 1.00$ |

## 4. Results

### 4.1. Determining the Inequality Index of VWT between China and Kazakhstan

4.1.1. TiVA Transfer between China and Kazakhstan

Before 1995, China was a net exporter of TiVA, gaining relatively more economic benefits, but after 1995, Kazakhstan took China's advantageous position. In detail, due to political chaos in Kazakhstan, the two countries experienced more trade fluctuations before 2001. After 2001, fluctuation dwindled, and Kazakhstan's export of value added to China kept increasing from USD 827.5 million to USD 3.6953 billion, gradually making China its key trade partner. As for China, its trade export to Kazakhstan stayed basically steady and stable with only a few fluctuations, with an average value of USD 813.9 million (Figure 4).

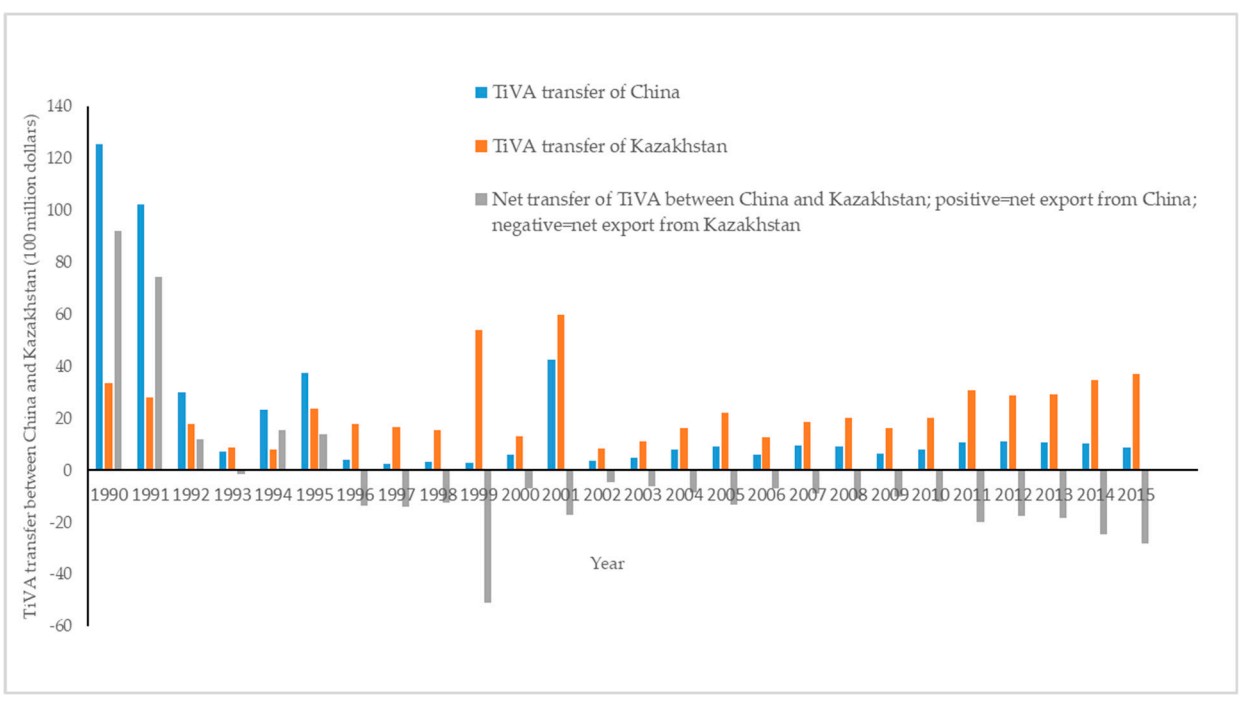

**Figure 4.** Transferred TiVA between China and Kazakhstan.

4.1.2. VWT between China and Kazakhstan

This study collects data on VWT between China and Kazakhstan during 1990–2015 from Eora (see Figure 5). In general, during 1990–1995, China was a net exporter and transferred more virtual water to Kazakhstan, with the output reaching the peak of 42.11 billion m$^3$ in 1991. After 1995, Kazakhstan became a net exporter, with the highest net output reaching 8.16 billion m$^3$ in 1999.

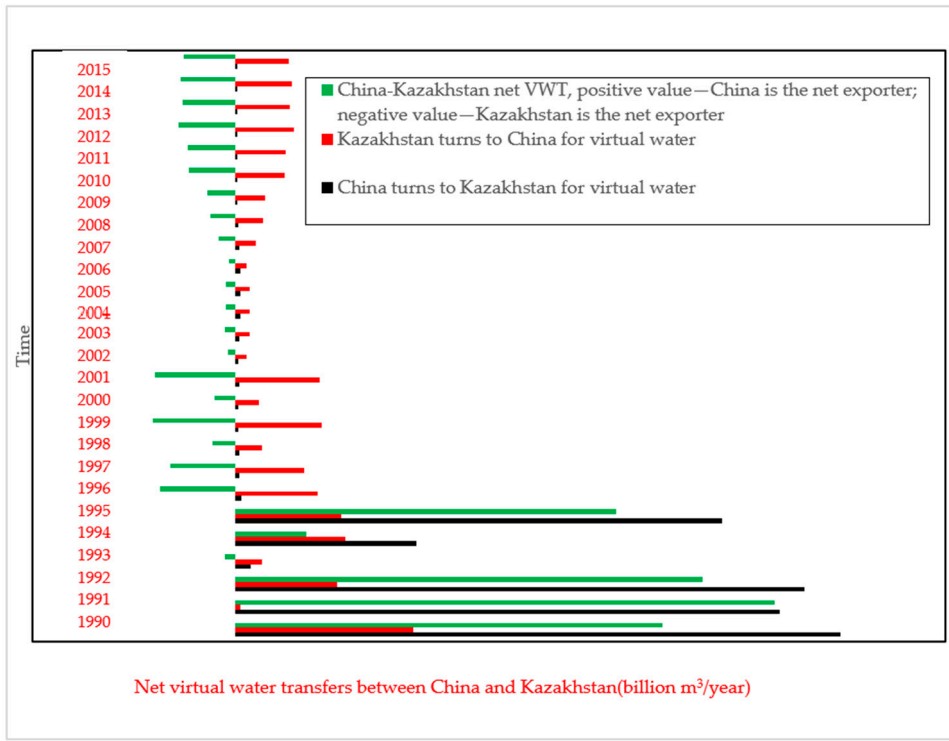

**Figure 5.** 1990–2015 comparison of VWT between China and Kazakhstan.

4.1.3. VWI Inequality Index of VWT in China and Kazakhstan

Based on the calculation, the Inequality Index of VWT between China and Kazakhstan stayed around 0.368 after 1995, with nearly zero change (Table 3 and Figure 6). Therefore, VWT between the two countries stayed stable in terms of inequality. From 1990–2015, China became a net importer as well as a beneficiary of virtual water since 1996, and Kazakhstan a net exporter of TiVA and a beneficiary of economic benefits. In addition, the inequality dot of VWT between the two countries is above the fair-trade line, indicating that the economic benefit that Kazakhstan receives by exporting each unit of virtual water to China are above the global average level. In other words, China drives the economic growth of Kazakhstan, and thus Kazakhstan enjoys relatively more economic benefits.

**Table 3.** Inequality Index and benefits of China and Kazakhstan.

| Year | Beneficiary of Virtual Water (Net Importer of Virtual Water) | Beneficiary of Economy (Net Exporter of TiVA) | VWI ($\times 10^{-2}$) |
|---|---|---|---|
| 1990 | Kazakhstan | China | 61.4971 |
| 1991 | Kazakhstan | China | 39.8408 |
| 1992 | Kazakhstan | China | 62.3452 |
| 1993 | China | Kazakhstan | 36.7883 |
| 1994 | Kazakhstan | China | 87.7068 |
| 1995 | Kazakhstan | China | 36.9415 |
| 1996 | China | Kazakhstan | 36.9415 |
| 1997 | China | Kazakhstan | 36.8217 |
| 1998 | China | Kazakhstan | 36.7995 |
| 1999 | China | Kazakhstan | 36.7915 |
| 2000 | China | Kazakhstan | 36.7879 |
| 2001 | China | Kazakhstan | 36.7984 |
| 2002 | China | Kazakhstan | 36.7960 |
| 2003 | China | Kazakhstan | 36.7975 |
| 2004 | China | Kazakhstan | 36.7978 |
| 2005 | China | Kazakhstan | 36.7962 |

**Table 3.** *Cont.*

| Year | Beneficiary of Virtual Water (Net Importer of Virtual Water) | Beneficiary of Economy (Net Exporter of TiVA) | VWI ($\times 10^{-2}$) |
|---|---|---|---|
| 2006 | China | Kazakhstan | 36.8006 |
| 2007 | China | Kazakhstan | 36.7902 |
| 2008 | China | Kazakhstan | 36.7901 |
| 2009 | China | Kazakhstan | 36.7902 |
| 2010 | China | Kazakhstan | 36.7902 |
| 2011 | China | Kazakhstan | 36.7896 |
| 2012 | China | Kazakhstan | 36.7903 |
| 2013 | China | Kazakhstan | 36.7903 |
| 2014 | China | Kazakhstan | 36.7901 |
| 2015 | China | Kazakhstan | 36.7902 |

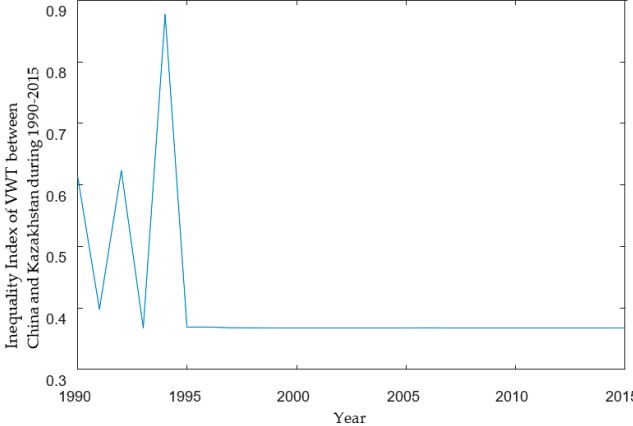

**Figure 6.** Inequality Index of VWT between China and Kazakhstan.

### 4.2. WSI in the Riparian Zones of Ili River between China and Kazakhstan

Via methods introduced in Section 3.2, this study obtained WSI in the riparian zones of the Ili River between China and Kazakhstan from 1990–2015. Additionally, to better categorize, this study sets every five years as a period for calculation. The results are shown in Figure 7.

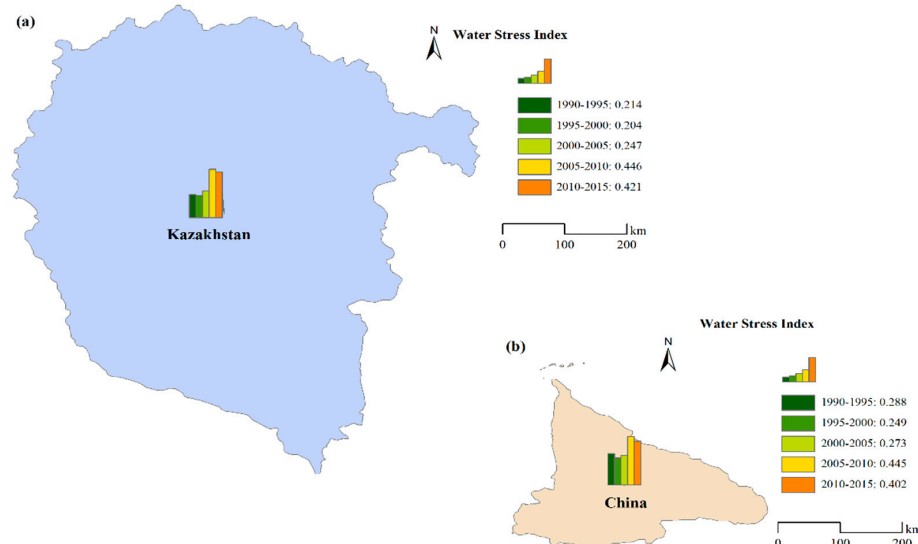

**Figure 7.** (**a,b**) Spatiotemporal evolution of WSI in the riparian zones of Ili River between China and Kazakhstan during 1990–2015.

According to Mekonnen et al. [57], WSI in the riparian zones of the Ili River generally ranges from 0.25 to 0.5, so both countries suffer from medium water stress. In addition, according to Figure 7, water stress in the study area showed an overall upward trend since 1990—water stress in Kazakhstan rose from 0.214 to 0.421, and that in China from 0.288 to 0.402.

*4.3. Types for China and Kazakhstan*

Based on the results in Section 4.1.3, since 1996, the Inequality Index of VWT in China and Kazakhstan fluctuated around 0.368 (0.368 < 0.5), indicating that VWT between the two countries was highly inequal, and China was at a disadvantage while Kazakhstan had the upper hand. According to Section 4.2, the minimum and maximum values of water stress in the riparian zones of China and Kazakhstan are (0.288, 0.445) and (0.214, 0.446), respectively, both at medium scarcity. Therefore, China and Kazakhstan both fall into type 2.

## 5. Discussion

*5.1. Analysis on Management Policies on Water Quantity Conflict in Transboundary Rivers Embedded with Virtual Water*

5.1.1. Current Management Policies

At present, the policies that countries within a drainage basin adopt to deal with the water quantity issues include: (1) Making fair and reasonable allocation plans of water rights (transboundary river's physical water resources)—key to solving water quantity conflict. It is the top choice of most transboundary rivers have been trying to solve the conflict by making plans to allocate water rights [14–23]. (2) Meeting each other's demand based on mutual benefit and win–win principle. It can be an exchange of homogenous rights—water rights for water rights, or an exchange of heterogenous rights—rights and interests of flood control for pecuniary benefits [70]. (3) Alleviating supply–demand imbalance by exchanging water rights [24–29]. Water rights trading is a new concept and strategy targeted at water quantity conflict in transboundary rivers. To be specific, water rights trading is designed to, based on the initial allocation of water quantity, redistribute the water of transboundary rivers and optimize and better allocate water resources, so as to solve the spatiotemporal mismatch in economic growth and water resources of countries within the same drainage basin.

5.1.2. Analysis on Management Policies in Different Types

Countries within a drainage basin are devoted to reducing the water quantity conflict in transboundary rivers, to achieve the "community with a shared future for water security." Based on the current management policies and analysis in Sections 3.3 and 5.1.1, this study analyzes the preferential policies in each type to further alleviate or reduce water quantity conflict.

(1) Type 1–3 share the feature of high inequality. Therefore, the transboundary river's physical water resources initial water rights allocation must include VWT and convert and embed it into the allocated water. However, due to difference in water stress as well as relative gains, countries have different emphases and expectations.

a. A country has low water stress and has an advantageous position. When allocating water, low water stress gives the country a lower expectation, and more wiggle room to negotiate for more quantifiable water. If a country is at a disadvantage, it means, when the country chooses between asking for more physical water and benefits in other aspects during water quantity allocation, it prefers exchanging for more benefits with other countries.

b. A country has medium water stress and has an advantageous position. When allocating water quantity with other countries, the country can moderately reduce its expectation and make decisions based on its benefits and relatively severe water stress. First, it can exchange with others for benefits, to reverse the outward transfer of its physical water resources. Second, it can proactively trade water rights with others, to avoid spatiotemporal

mismatch in water resources. If the country is at a disadvantage, it means the country, when converting and embedding VWT into quantifiable water during water quantity allocation, can ask other countries for more water through friendly negotiation.

c. A country has high water stress and has an advantageous position. In other words, the country should proactively negotiate with others, to maintain its original share of physical water by means of exchanging benefits in other aspects. If other countries have higher water stress and ask for more water, then the country can, based on the principal of being fair and reasonable, expand wiggle room, and refuse to step back. If it is at a disadvantage, then the country can, while allocating water quantity, ask other countries for more wiggle room in water quantity negotiation through friendly negotiation.

(2) Type 4–6 share the feature of low inequality. Therefore, the transboundary river's physical water resources initial water rights allocation can neglect the impact from VWT. In addition, countries within the same drainage basin can implement management policies with targeted emphases in light of water stress.

a. A country with low water stress can negotiate water allocation plans based on the principles of being fair, reasonable, and sustainable. In addition, countries can trade water rights and improve the water-use efficiency to reach a win–win result.

b. A country with medium water stress can ask to allocate physical water based on each country's contribution to and usage of physical water. Moreover, the country can exchange benefits in other aspects with other countries, to proactively seek water rights trading, and reduce spatiotemporal mismatch of water resources.

c. A country with high water stress can, besides making sure of its water rights, exchange benefits in other aspects with other countries, and trade water rights, to ensure water security for the riparian zones of each country.

*5.2. Analysis of Management Policies on Water Quantity Conflict in China–Kazakhstan Ili River Embedded with Virtual Water*

According to Sections 4.3 and 5.1, China and Kazakhstan fall into Type 2. Therefore, on the one hand, Kazakhstan has a great advantage, and the Xinjiang Uygur Autonomous Region of China contributes more to the Ili River. On such basis, Kazakhstan can lower its expectation and actively ask for exchanging benefits to reverse the outward transfer of its physical water. China, in turn, can ask for more water quantity in a friendly way. On the other hand, as the economy in the riparian zones of the Ili River between China and Kazakhstan thrives and runoff water changes, the initial plan of allocating water rights is hard to adjust to the changing needs for the water and spatiotemporal mismatch of the two countries, nor to improve the water-use efficiency, which could lead to water quantity conflict. As a result, water rights trading, as a fresh idea in solving the water quantity conflict of the Ili River, can be accepted. It is applicable to producing mutual benefit results between China and Kazakhstan, and it helps achieve the Pareto optimality of utilizing the water resources in the riparian zones of the Ili River.

In addition, China and Kazakhstan should negotiate their water-use objectives on the basis of respecting each other's benefits. Because one critical trigger of water quantity conflict in transboundary rivers is that a country only focuses on its own benefits yet ignores other countries' benefits, especially in agriculture where water is a critical input. Hence, the riparian zones of the two countries must facilitate an intensive economy and develop water-saving technologies and the recycling of water resources, in order to transform and upgrade foreign trade, reduce the intensity of water use in each industry, especially agriculture, and adjust the production structure of intermediate products. Furthermore, the two countries ought to import more water-consuming products and increase the value added to agricultural products, and export less virtual water. In doing so, China and Kazakhstan can, based on securing the basic needs for water, restructure each other's economy, enjoy a more complementary economic structures, achieve "regional economic integration", and realize both mutual development and water conservation.

## 6. Conclusions

Relevant studies ignore the invisible impact that VWT embedded in the commodity trade of countries within the same drainage basin has on water quantity conflict in transboundary rivers, which would make management policies less fair and reasonable. Therefore, this study first measures the VWT and transferred TiVA between China and Kazakhstan based on the EE-MIRO model and constructs the Inequality Index of VWT. Next, based on WSI model, and data from NASA, ESA and relevant literature, this study calculates the water stress of the riparian zones of the Ili River. Then, different types are founded to optimize and make management policies targeted at each type. Last, with the Ili River as the case, this study draws the following conclusions:

(1) During 1990–1995, China was basically a net exporter of virtual water and TiVA; after 1995, Kazakhstan took that position.

(2) Since 1996, the Inequality Index of VWT between the two countries stayed around 0.368 (0.368 < 0.5), indicating a relatively high inequality in VWT, and that China was at a disadvantage while Kazakhstan had the upper hand.

(3) During the study period, the minimum and maximum values of water stress in the riparian zones of the Ili River between China and Kazakhstan are (0.288, 0.445) and (0.214, 0.446), respectively, both at medium scarcity.

(4) China and Kazakhstan both fall into Type 2, and Kazakhstan has an advantageous position. So, while allocating the water quantity of the Ili River, Kazakhstan can lower its expectation and proactively ask for exchanging benefits in other aspects and reduce outward transfer of its physical water. In addition, the two countries can seek suitable ways of water rights trading.

**Author Contributions:** Writing, X.X.; Providing case and idea, F.W.; Providing revised advice, Q.Y., Y.Z., X.C. All authors have read and agreed to the published version of the manuscript.

**Funding:** This paper was supported by the Major Projects of the National Social Science Fund of the People's Republic of China (No. 17ZDA064). Anhui Provincial Education Department Humanities Key Fund (No.SK2021A0652). Postgraduate Research & Practice Innovation Program of Jiangsu Province (KYCX21_0446). Tongling College Talent Fund (2021tlxyr15). The Ministry of Education of Humanities and Social Science Project (No. 21YJCZH206).

**Institutional Review Board Statement:** Not applicable.

**Informed Consent Statement:** Not applicable.

**Data Availability Statement:** Not applicable.

**Conflicts of Interest:** The authors declare no conflict of interest.

## Abbreviations

| | |
|---|---|
| Virtual water | (VW) |
| Virtual water transfer | (VWT) |
| Water stress index | (WSI) |
| Trade in value added | (TiVA) |
| Environmentally extended multiregional input–output | (EE-MRIO) |
| Critical ratio | (CR) |
| Falkenmark water stress indicator | (FWSI) |
| NASA | (National Aeronautics and Space Administration) |
| ESA | (European Space Agency) |
| The Eora Global Supply Chain Database | (EORA) |
| Global trade analysis project | (GATP) |
| World input–output database | (WIOD) |
| Inter-country input–output | (ICIO) |
| Inequality Index of VWT | (II) |

## Appendix A

**Table A1.** The modified EE-MRIO table.

| | | Output | Intermediate Use | | | | | | | | | | Final Demand | | | | Total Output |
|---|---|---|---|---|---|---|---|---|---|---|---|---|---|---|---|---|---|
| | | | Basin Country 1 | | | ... | Basin Country $(m-1)$ | | | Other Country $m$ | | | Basin Country 1 | ... | Basin Country $(m-1)$ | Other Country $m$ | |
| Iutput | | | Industry 1 | ... | Industry $n$ | ... | Industry 1 | ... | Industry $n$ | Industry 1 | ... | Industry $n$ | | | | | |
| Intermediate use | Basin country 1 | Industry 1 | $y_{11}^{11}$ | ... | $y_{1n}^{11}$ | ... | $y_{11}^{1(m-1)}$ | ... | $y_{1n}^{1(m-1)}$ | $y_{11}^{1m}$ | ... | $y_{1n}^{1m}$ | $f_1^{11}$ | ... | $f_1^{1(m-1)}$ | $f_1^{1m}$ | $y_1^1$ |
| | | ... | ... | ... | ... | ... | ... | ... | ... | ... | ... | ... | ... | ... | ... | ... | ... |
| | | Industry $n$ | $y_{n1}^{11}$ | ... | $y_{nn}^{11}$ | ... | $y_{n1}^{1(m-1)}$ | ...... | $y_{nn}^{1(m-1)}$ | $y_{n1}^{1m}$ | ... | $y_{nn}^{1m}$ | $f_n^{11}$ | ... | $f_n^{1(m-1)}$ | $f_n^{1m}$ | $y_n^1$ |
| | ... | ... | ... | ... | ... | ... | ... | ... | ... | ... | ... | ... | ... | ... | ... | ... | ... |
| | Basin country $(m-1)$ | Industry 1 | $y_{11}^{(m-1)1}$ | ... | $y_{1n}^{(m-1)1}$ | ... | $y_{11}^{(m-1)(m-1)}$ | ... | $y_{1n}^{(m-1)(m-1)}$ | $y_{11}^{(m-1)m}$ | ... | $y_{1n}^{(m-1)m}$ | $f_1^{(m-1)1}$ | ... | $f_1^{(m-1)(m-1)}$ | $f_1^{1m}$ | $y_1^{m-1}$ |
| | | ... | ... | ... | ... | ... | ... | ... | ... | ... | ... | ... | ... | ... | ... | ... | ... |
| | | Industry $n$ | $y_{n1}^{(m-1)1}$ | ... | $y_{nn}^{(m-1)1}$ | ... | $y_{n1}^{(m-1)(m-1)}$ | ... | $y_{nn}^{(m-1)(m-1)}$ | $y_{n1}^{(m-1)m}$ | ... | $y_{nn}^{(m-1)m}$ | $f_n^{(m-1)1}$ | ... | $f_n^{(m-1)(m-1)}$ | $f_n^{1m}$ | $y_n^{m-1}$ |
| | Other country $m$ | Industry 1 | $y_{11}^{m1}$ | ... | $y_{1n}^{m1}$ | ... | ... | ...... | ... | $y_{11}^{mm}$ | ... | $y_{1n}^{mm}$ | $f_1^{m1}$ | ... | $f_1^{m(m-1)}$ | $f_1^{mm}$ | $y_1^m$ |
| | | ... | ... | ... | ... | ... | ... | ... | ... | ... | ... | ... | ... | ... | ... | ... | ... |
| | | Industry $n$ | $y_{n1}^{m1}$ | ... | $y_{nn}^{m1}$ | ... | ... | ...... | ... | $y_{n1}^{mm}$ | ... | $y_{nn}^{mm}$ | $f_n^{m1}$ | $f_n^{m2}$ | $f_n^{m(m-1)}$ | $f_n^{mm}$ | $y_n^m$ |
| Added Value | | | $v_1^1$ | ... | $v_n^1$ | ... | ... | ...... | ... | $v_1^m$ | ... | $v_n^m$ | | | | | |
| Total input | | | $y_1^1$ | ... | $y_n^1$ | ... | ... | ...... | ... | $y_1^m$ | ... | $y_n^m$ | | | | | |
| Direct water input | | | $w_n^r$ | | | | | | | | | | | | | | |

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
