# Peer review of "Analysis on Management Policies on Water Quantity Conflict in Transboundary Rivers Embedded with Virtual Water—Using Ili River as the Case"

_sustainability, doi:10.3390/su14159406_

Round 1
Reviewer 1 Report
1、The full name of the abbreviation should be provided in the appendix. What’s more, add units to all variables.
2、It is suggested that the research structure in Section 2.1 be adjusted to the introduction section.
3、In the paper, the statements of figure are mixed (fig. and figure.), please make a unified expression. The formula also should be aligned, please revise it. For example, formula (1) and (2).
4、There are some minor spelling errors throughout the paper. Please double-check all the words and sentences.
Author Response
Dear reviewer:
We have finished the proof reading and checking carefully. Some corrections about the proof and the answers to the queries are provided in appendix.

Reviewer 2 Report
This study explored the inequality of virtual water transfer (VWT) between China and Kazakhstan and found that a relatively high inequality of VWT between the two countries while there are some problems which need to fix before get acceptance of this journal.
First, why choose Kazakhstan? As the authors used the global multi-regional input-output model (MRIO) like EORA which covered the most countries of the world, but why the authors choose the virtual water transfer between China and Kazakhstan? Why not other countries? What’s the specificity of Kazakhstan? The authors should make more explanation about this. Second, why focused on transboundary rivers? To be honest, I haven’t seen the significance of transboundary rivers for this study. The transboundary rivers most related with physical water resources transfer rather than virtual water. While the authors seem did not consider the physical water resources transfer. If the authors want to make a point about transboundary rivers, the physical water resources should be compared with VWT. Third, the authors established the inequality index which focused on quantity of water resources and trade. While the water scarcity should be included in the index. If more water resources with higher scarcity transferred through trade even achieving fair-trade as the authors show, there would still be inequality as the environmental impact of water consumption with different scarcity levels gets great variation. Fourth, the scenario set seems vague to me. To my knowledge, scenario set should be projection for future to show the impact of policy change, population growth or economic development. The scenario set in this study may be more appropriate taken as different classification or types? The authors should fix this. Fifth, the VWT should be the two countries, how to make the results related with Ili River? The authors seem emphasize the water stress of the Ili River, if the stress did not have connection with VWT then why bother? The authors should make more explanation about this.
Based on the above, this paper deserves a major revision before getting the publication of Sustainability.

Author Response
Dear Reviewer:
We really appreciate your work and highly appreciated the reviewer for your constructive comments and suggestions.
Based on the comments and suggestions, we carefully revised the manuscript. The details are in appendix

Round 2
Reviewer 2 Report
The authors had made revision to my suggestions. I have no further comments.